# Feasibility Study of Grinding Circulating Fluidized Bed Ash as Cement Admixture

**DOI:** 10.3390/ma15165610

**Published:** 2022-08-16

**Authors:** Xingquan Du, Zhong Huang, Yi Ding, Wei Xu, Man Zhang, Lubin Wei, Hairui Yang

**Affiliations:** 1School of Chemical and Environmental Engineering, China University of Mining and Technology (Beijing), Beijing 100083, China; 2Key Laboratory of Thermal Science and Power Engineering, Department of Energy and Power Engineering, Tsinghua University, Ministry of Education, Beijing 100084, China; 3SDIC Power Holding Co., Ltd., Beijing 100034, China; 4SDIC Panjiang Electric Power Co., Ltd., Liupanshui 553529, China

**Keywords:** circulating fluidized bed ash, cement admixture, grinding, water-demand ratio, activity index

## Abstract

With the widespread application of circulating fluidized bed (CFB) combustion technology, the popularity of CFB ash (CFBA) has increased dramatically and its production and large-scale utilization have become increasingly important. In the context of carbon neutrality peaking, using CFBA as a cement admixture as an effective method of resource utilization not only reduces the pressures caused by carbon emissions in the cement industry but also solves the environmental problems caused by CFBA depositing. However, the formation conditions of CFBA are worse than those of traditional pulverized coal boilers. CFB ash is the combustion product of coal at 850 °C–950 °C, and the characteristics of CFBA usually include a loose and porous structure with many amorphous substances. Furthermore, it has the disadvantages of large particle size, high water-demand ratio, and low activity index when it is directly used as a cement admixture. In this study, CFBA (including fly ash (CFBFA) and bottom ash (CFBBA)) produced by a CFB boiler without furnace desulfurization with limestone was used as a cement admixture material, and the effect of grinding on the fineness, water-demand ratio, and activity index of CFBA were studied. The experimental results showed that the grinding effect could significantly reduce the fineness and water-demand ratio of CFBA as a cement mixture and improve the activity index. With the increase in the grinding time, the water-demand ratio of CFBA first decreased and then increased. CFBBA ground for 10 min and CFBFA ground for 4 min can reduce the water-demand ratio of CFBA by up to 105% and increase the compressive strength of 28-day-old CFBA cement by 7.05%. The grinding process can ensure that CFBA meets the Chinese standards for a cement admixture and realize the resource utilization of CFBA.

## 1. Introduction

Circulating fluidized bed (CFB) combustion technology is widely used because of its good fuel flexibility, low pollution emissions, and large turndown ratio [1,2,3]. A large amount of ash is discharged during the boiler operation because low-rank fuels are used commonly in CFB boilers. In 2021, 300 million tons of CFB ash (CFBA) were discharged in China, which caused problems such as low utilization rates [4], changes in the occupancy of farmland [5,6], and environmental pollution [7,8,9]. In the face of such problems, the Chinese government has issued relevant policies requiring CFB power plants to determine their power generation according to their ash treatment capacity. Therefore, it is urgent to develop CFBA resource utilization technology.

CFB combustion technology was developed after pulverized coal (PC) boilers so the research on ash utilization is lagging. CFB ash utilization has been studied in the literature in recent years including its use as a fine aggregate in compacted concrete [10], as a geopolymer [11,12], as a cement admixture [13,14,15,16,17], in road construction materials [18], and in metal recycling [19,20]. Among them, the best large-scale utilization of CFB ash is in the production of cement admixture materials [21]. Cement production in China amounted to 2.38 billion tons in 2021 [22], and the pressure to reduce carbon in the cement industry is huge in the context of carbon neutrality peaking. The use of CFBA as a cement admixture not only solves the environmental pollution problem caused by CFBA depositing but also reduces the output of cement clinker, thereby reducing carbon emissions, which has received widespread attention from scholars in recent years.

However, CFB ash is the combustion product of coal at 850 °C–950 °C [23], and the characteristics of CFBA include its loose and porous structure with many amorphous substances [24]. Normally, limestone is fed into the furnace during the CFB boiler operation in order to control the SO_2_ emission. The CaO decomposed by the limestone in the boiler reacts with SO_2_, thus reducing the SO_2_ content in the gas emission and ensuring that the SO_2_ concentration in the exhaust gas meets the national requirements, which is of great significance to the control of pollutants [25,26]. This results in a relatively high content of sulfur and free CaO in the ash. The free CaO restricts its application as a cement admixture and creates new challenges for the utilization of CFBA [27,28,29].

Wu et al. stated that CFBC fly ash has the potential to replace cementitious materials and is a substitute for pozzolan [13]. Li et al. compared desulfurized ash and fly ash and found that by adding a 1% modifier (a kind of alkali-activated activator) and a 0.8% modifier (the mixture of naphthalene-based water reducer and an adjustable solidification agent), the strength activity index of desulfurized ash increased by 23–37% and 15–28.8%, respectively [14].

Tkaczewska, E. used different admixtures of sulfonated melamine-formaldehyde condensate, sulfonated naphthalene-formaldehyde condensate, polycarboxylate, and polycarboxylate ether, resulting in a 15%, 31%, 42%, and 47% reduction in the water-demand ratio of fly ash [30].

Zheng et al. conducted a study under the condition of a water/cement (W/C) ratio of 0.35 and an amount of polycarboxylate superplasticizer of 0.2% by weight of cementitious materials was added to the mixture. The water-demand ratio was effectively reduced but the mortar intensity also decreased [31].

Sun et al. considered the strength, expansion, and void ratio for optimizing the ratio of the CFB ash–anhydrite–clinker so that CFB ash could be used as the source of the aluminosilicate material for supersulfated cement, thereby reducing carbon emissions [32].

CFBA contains more SO_3_ and free CaO, which limits its application. Even if power plants adopt flue gas desulfurization technology outside the boiler to reduce the content of sulfur and free CaO in CFBA, there are still problems such as a large particle size, large water-demand ratio, and low activity index when CFBA is directly used as a cement admixture. Grinding can reduce the problems of the particle size and water-demand ratio of CFBA [7,33,34,35] and improve the strength of the mortar to meet Chinese standards as a cement admixture. However, the key influencing factors in the quality of ground ash as a cement admixture still need to be investigated.

In this study, the focus is on the problems of the large particle size, large water-demand ratio, and low activity index of the CFBA produced by a 300 MW CFB boiler with external flue gas desulfurization technology in order to ensure that CFBA meets the Chinese standards for its use as a cement admixture. Furthermore, the effects of grinding on the fineness, water-demand ratio, and activity index of CFBA were studied, and the results provide new ideas for carbon reduction and CFB ash utilization in the cement industry.

## 2. Materials and Methods

### 2.1. Materials

The CFBA used in the test was from the 300 MW CFB boiler of Guotou Panjiang Power Co., Ltd. (Liupanshui, China) and included CFB fly ash (CFBFA) and CFB bottom ash (CFBBA). The cement was PI42.5 Portland cement and was sourced from China Fushun Cement Co., Ltd. (Fushun, China) Standard sand was standard sand produced by Xiamen ISO Standard Sand Co., Ltd. (Xiamen, China). The water was deionized.

### 2.2. Methods

In order to ensure that CFBA meets the requirements of cement admixtures found in Chinese standard GB-T 1596-2017, the required tests of CFBA in this study included fineness, loss of ignition, water content, total SO_3_ mass fraction, free CaO mass fraction, SiO_2_, Al_2_O_3_ and Fe_2_O_3_ total mass fraction, density, water-demand ratio, and strength activity index. Each experiment was measured at least twice and for some important research data (water-demand ratio and mild activity index), each experiment was measured at least three times.

The experimental process is shown in Figure 1. First, the CFBBA was screened through a 2.2 mm sieve and the large particles on the sieve were crushed to less than 2.2 mm with a KERP-180 × 150B sealing hammer crusher produced by Zhenjiang Kerui Sample Preparation Equipment Co., Ltd. (Zhenjing, China).

An XQM-4 vertical planetary mill produced by Changsha Tianchuang Powder Technology Co., Ltd. (Changsha, China) was used for grinding. The CFBFA and CFBBA were ground separately. The mass of the ash sample was 100 g per grinding, the speed of the ball mill was 300 r/min, and the variable condition was the grinding time. After the grinding, the ash sample was sieved through a sieve with a pore diameter of 0.9 mm, and the ash sample under the sieve was tested and used as a cement mixing material.

The SEM detection of CFBA was carried out with a Merlin Compact type Scanning Electron Microscope produced by Carl Zeiss AG (Oberkochen, Germany) under the conditions of an accelerating voltage of 0.05–30 kV and a resolution of 1.0 nm/15 kV.

The particle size and specific surface area of CFBA were measured under SOP using a Mastersizer 2000 laser particle sizer produced by Malvern Instruments Ltd. (Malvern, UK). The compound composition of CFBA was detected using an ARL PERFORM X-ray fluorescence spectrometer produced by Thermo Fisher (Shanghai, China). The operating technical indicators were target material: Rh; light tube power: 4 KW; excitation voltage: 60 kV (max); and excitation current: 140 mA (max). Equipped with 9 dichroic crystals, the element test range was B-U. The density of CFBA was measured according to Archimedes’ principle using Lee’s bottles and kerosene. Its value is the ratio of the weight of the ash residue to the increased volume of kerosene. The fineness of CFBA refers to the percentage of total mass on the screen after the CFBA was screened using an FYS-150B cement fineness negative pressure sieve analyzer (sieve hole is 45 μm) produced by Hebei Xinglan Construction Instrument Co., Ltd. (Cangzhou, China).

The loss of ignition (LOI) of CFBA was detected using an AS-1700 muffle furnace produced by Zhengzhou Ansheng Scientific Instrument Co., Ltd. (Zhengzhou, China). The method involved placing the CFBA into the muffle furnace, gradually increasing the temperature starting at room temperature, burning at 950 ± 25 °C for 1 h, and then placing the CFBA into the desiccator to cool it down to room temperature; the percentage of the mass CFBA lost to the total mass is the LOI.

The method for determining the water-demand ratio of CFBA was as follows. After the contrasting mortar and test mortar were stirred evenly under certain conditions according to the material ratio shown in Table 1, a JJ-5 type cement sand mixer produced by Hebei Xinglan Construction Instrument Co., Ltd. (Cangzhou, China) was used for stirring at low speed for 30 s. Sand was then added evenly at the beginning of the second 30 s while mixing at high speed, then the sample was left standing for 90 s, and finally, another 60 s of high-speed mixing was carried out. The cement mortar sample was vibrated 25 times using an NLD-3 testing instrument produced by Hebei Xinglan Construction Instrument Co., Ltd. (Cangzhou, China). The average diameter of the mortar in both vertical directions was measured. When the average value of both tests reached 180 mm, the water-demand ratio of the test mortar(W2) and contrast mortar(W1) were the water-demand ratio of the CFBA.

The CFBA intensity activity index was measured as follows. First, according to the composition shown in Table 2, the material was stirred evenly and then the mixing method was the same as in the measurement of the water-demand ratio mentioned above. The mortar was put into the mold and vibrated 60 times, and then the mortar was placed into an environment of 20 ± 1 °C with a humidity greater than 90% for 24 h. The test mold was then removed and placed into a 20 ± 1 °C water cure to a fixed age. The compressive strength was tested using a YAW-300 microcomputer-controlled electro-hydraulic servo pressure testing machine produced by Jinan Time Shijin Testing Machine Co., Ltd. (Jinan, China).The flexural strength was tested using a DKZ-5000 electric flexural tester produced by Wuxi Jianyi Instrument and Machinery Co., Ltd. (Wuxi, China). The activity index is the ratio of compressive strength of test mortar to contrast mortar.

## 3. Results and Discussion

### 3.1. Physicochemical Analysis of CFBA 

#### 3.1.1. Chemical Composition of CFBA

Table 3 lists the results of the XRF analysis of CFBA’s components. The boiler co-fires coal slurry and coal gangue, the coal slurry particle size is fine, and the combustion process produces ash mainly in the form of CFBFA discharged from the boiler, whereas the gangue particle size and density are relatively larger, the combustion produces ash mainly in the form of CFBBA discharged from the boiler. The test results show that the compositions of the fly ash and bottom ash are different because the mineral compositions of the coal slurry and gangue are different. For example, there are high SiO_2_ and CaO contents in boiler CFBFA and high Fe_2_O_3_ content in CFBBA. There are a lot of active SiO_2_ and Al_2_O_3_ contents in CFBA [34], which have high pozzolanic activity and can react with the Ca(OH)_2_ generated in cement to generate gel products such as CaO–SiO_2_–H_2_O, CaO–Al_2_O_3_–H_2_O, etc. These gel products provide strength for the mortar. Flue gas desulfurization technology is used outside the boiler and the limestone is not placed into the furnace during the boiler operation so the contents of CaO and SO_3_ in CFBA are lower than those in traditional CFBA and free CaO is rarely almost negligible. This is more beneficial to the use of CFBA as a cement admixture.

#### 3.1.2. Physicochemical Characteristics of CFBA

Studies have shown that CFBA needs to meet the requirements of second-class pulverized coal ash for use in a cement admixture according to standard GB-T 1596-2017. The fineness, water-demand ratio, LOI, moisture content, sulfur trioxide, density, and activity index of CFBFA and CFBBA were measured, respectively, during the experiment. As listed in Table 4, the density and size of the gangue are bigger than that of coal slurry and it takes a longer residence time to burn in the boiler so the LOI of CFBBA is lower than that of CFBFA and the density is larger than that of CFBFA. However, the CFB boiler combustion temperature is relatively lower and since it cannot melt most minerals, most of the minerals in CFBA remain in their original shape and do not melt into spherical particles. This results in the loose and porous structure of CFBA(Figure 2) and the water-demand ratio is high. The other parameters meeting the Chinese standard requirements of second-class pulverized coal ash according to standard GB-T 1596-2017 include the LOI, moisture content, sulfur trioxide, free CaO, and the density of CFBBA and CFBFA. Only the fineness, water-demand ratio, and activity index of CFBBA do not meet the requirements. CFBFA only has two parameters that do not meet the requirements. One is the water-demand ratio and the other is the activity index. In this study, the grinding process was used to treat CFBA in order to meet the fineness requirements. In addition, the influence of CFBA particle size on the water-demand ratio and mortar strength was also studied.

### 3.2. Influence of Grinding Time on the Properties of CFBA

#### 3.2.1. Influence of Grinding Time on CFBBA

The results of the fineness of CFBBA after different grinding times are shown in Figure 3. With increased grinding time, the fineness of CFBBA became smaller and smaller but the change gradually became slower. The particles larger than 45 μm were already crushed by grinding for 6 min, which led to a sharp decrease in the 45 μm particle residue above the sieve at the beginning of the grinding. After a grinding time of more than 6 min, although the proportion of particles larger than 45 μm continued to decrease, the decrease rate was smaller. This is because the CFBBA particles were larger in the early stage of grinding and were broken into fine particles affected by the impact force of the steel balls. With the decrease in the particle size of CFBBA, the particle size became much smaller than the diameter of the steel balls in the ball mill. The force on the ash particles was mainly from the grinding force between the steel balls and the tank or the steel balls in the ball mill, which made the CFBBA particle size smaller. The impact force on the ash particles was relieved because the larger particles were mixed with fine particles so the crushing of the coarse particles became more difficult. In fact, when the grinding time reached 5 min, the fineness of CFBBA had already met the condition of being less than 30%, which was in line with standard GB/T 1596-2017.

The SEM images of CFBBA after different grinding times are shown in Figure 4 and the particle size distribution is shown in Figure 5 (the numbers in the legend represent the grinding time (min)). The particle sizes corresponding to 10%, 50%, and 90% of the cumulative particle size distribution (d(0.1), d(0.5), and d(0.9)) are also listed in Table 5. The specific surface area at different grinding times is shown in Figure 6. It can be seen that with the increase in the grinding time, the CFBA particles gradually decreased. When the grinding time was 20 min, the inter-particle attraction was larger and the fine particles were wrapped around the large particles to form agglomeration. When the grinding time was less than 18 min, with the increase in the grinding time, the particle size distribution of CFBBA gradually shifted to the left, the specific surface area gradually increased, and the particle sizes d(0.1), d(0.5), and d(0.9) gradually became smaller. The changing trends in the three parameters became slower and slower and this tendency was consistent with the fineness results seen in Figure 3. When the grinding time reached 20 min, the d(0.5) and d(0.9) of the CFBBA particles increased and the specific surface area decreased correspondingly.

The main reason for the above results is that there were fracture intersections of lattice destruction in the fine particles, which made the surface of the fine particles have a large number of electric charges and the particles were attracted to each other by electrostatic force and van der Waals force resulting in agglomeration. The particles’ agglomeration caused the large particles to be wrapped in the fine particles. When the steel balls ground the particles, the force between the particles had to be overcome to continue grinding, which not only reduced the grinding efficiency but also increased the grinding energy consumption.

#### 3.2.2. Effect of Grinding Time on the Water-Demand Ratio of CFBA

In the actual operation of the CFB boiler, the mass ratio of CFBFA and CFBBA is 7:3. In order to realize the full utilization of CFBA, in the experiment on the water-demand ratio, CFBFA and CFBBA were ground and mixed into cement according to this ratio to measure the water-demand ratio. The water-demand ratio method was introduced in Section 2.2 in line with standard GB/T 1596-2017 and the results are as follows.

Because the fineness of CFBFA can meet the requirements of standard GB/T 1596-2017 even without grinding, only CFBBA needed to be ground. The water-demand ratio of CFBA with grinding time is shown in Figure 7. It can be seen that the water-demand ratio of CFBA first decreased and then increased with the increase in the grinding time of CFBBA, but it still did not reach the 105% requirement stated in standard GB/T 1596-2017.

Many factors affect the water-demand ratio of CFBA including the chemical composition, carbon content, particle size, and particle shape of CFBA. The chemical composition of CFBA in this experiment was fixed so its mechanism can be explained by three other aspects. In Figure 8 and Figure 9, orange represents the CFBA particles and blue represents water.

(a)As shown in Figure 8, with the decrease in the particle size, some of the particles played a micro-filling role, and the fine particles filled in the gaps in the coarse particles, reducing the porosity and the use of water [36]. Furthermore, with the addition of CFBA to the cement, the particle size distribution of the CFBA cement system changed, which affected the bulk density of the slurry by reducing the amount of retained water [37,38]. In addition, the grinding effect made the particles of CFBA more and more spherical, and the fine spherical particles also played a role in the lubrication of the balls. Therefore, less water was needed for the mortar to flow better, thereby reducing the water-demand ratio of CFBA.(b)As shown in Figure 9, when all particles were reduced, pores were generated between the particles and the porosity increased, which worsened the fluidity of the mortar [39]. Therefore, more water was needed for the mortar to achieve a better flow, resulting in an increase in the water-demand ratio of CFBA. This explains the mechanism by which the water-demand ratio increased when the grinding time further increased.(c)The effect of the carbon content of CFBA on the water-demand ratio was also tested. Wu Bin et al. showed that the water-demand ratio of pulverized coal ash increased with the increase in the carbon content [40]. Similarly, the greater the burn losses of CFBA, the greater the carbon content. The particles with high carbon content were coarse, porous, non-gelling, and easily absorb water, which increased the water-demand ratio of CFBA.

For the first test, with the increase in the CFBBA grinding time, the water-demand ratio of CFBA first decreased and then increased because (a) and (b) played a major role. The reason that the minimum value could reach 105% was that only CFBBA was ground, whereas CFBFA was not, and the carbon content of CFBA is high. So, a second experiment was needed.

In the second experiment, the grinding time of CFBBA was fixed at 10 min, and then CFBFA was ground at different times. The results of the water-demand ratio of CFBA are shown in Figure 10. It can be seen that when the grinding time of CFBFA reached 4 min, the water-demand ratio of CFBA reached 105% as required in standard GB/T 1596-2017. In addition, when the grinding time of CFBFA reached 7 min and 8 min, the water-demand ratio of CFBA decreased to 104%. The smaller the water-demand ratio of CFBA, the higher the proportion of CFBA used in the cement admixture and the higher the engineering utilization value. When the grinding time of CFBFA increased, the water-demand ratio of CFBA increased again.

In this experiment based on a mixing ratio of CFBFA to CFBBA of 7:3, considering the cost of grinding, “the grinding time of CFBFA is 4 min, and the grinding time of CFBBA is 10 min” were the best grinding times for meeting the water-demand ratio of the CFBA conditions.

#### 3.2.3. Effect of Grinding Time on Compressive Strength and Flexural Strength of CFBA Mortar

The compressive and flexural strengths of the CFBA mortar were tested according to the GB/T 1596-2017 standard. Similar to the test for the water-demand ratio, CFBFA: CFBBA was mixed into the cement at a ratio of 7:3 to test the compressive and flexural strengths of the mortar. The experimental method was described in Section 2.2. In the test, CFBFA was ground for 4 min, CFBBA was ground at different times, and then CFBFA and CFBBA were mixed with cement to make mortar. The compressive and flexural strengths of the test mortar and the compared mortar after 3 d, 7 d, and 28 d are shown in Figure 11 and Figure 12. With the increase in the grinding time, the compressive and flexural strengths of CFBA initially increased rapidly. After the grinding time reached 10 min, the compressive and flexural strengths of the mortar hardly changed even when the grinding time increased further. Unground CFBBA was too large to be used. In this experiment, the CFBBA at a grinding time of 2 min was used as a reference and it was found that when the CFBBA was ground for 10 min, the compressive strength of the 28-day-old mortar of CFBA increased by 7.05%. The specific surface area of CFBBA increased due to grinding, allowing the active substances such as SiO_2_ and Al_2_O_3_ in CFBA to have a larger contact area to react with the cement, which sped up the reaction rate and improved the compressive and flexural strengths of the mortar. When the grinding time of CFBBA reached a certain condition, the reaction rate did not increase after reaching the maximum. Therefore, grinding for a certain period of time can promote the strength of CFBA cement admixture mortar, which is consistent with the conclusions of previous experiments [41].

The relationship between the strength activity indexes of CFBA at 3 d, 7 d, and 28 d with the grinding times of CFBBA is shown in Figure 13. The strength activity coefficient of the mortar mixed with CFBA was lower at 3 d, whereas the strength activity coefficient at 7 d was relatively good and the strength activity coefficient at 28 d was as high as 90%. When the grinding time of CFBBA was 12 min, the intensity activity coefficient of CFBA at 28 d was the highest, reaching 95.9%, which far exceeded the 70% required by the GB/T 1596-2017 standard. When the grinding time of CFBBA was not less than 10 min, the water-demand ratio met the requirements. At this time, the strength activity coefficient of the mortar mixed with CFBA also met the 70% requirement of the GB/T 1596-2017 standard.

## 4. Conclusions

In this study, the ash from a CFB boiler without in-furnace desulfurization used as a cement admixture was studied. Most of the physical and chemical properties of CFBA met the applicable Chinese standards of second-class pulverized coal ash, the contents of SiO_2_ and Al_2_O_3_ in CFBA were even higher, and the sulfur and free CaO were lower. Because three parameters did not meet the standards, the grinding process was used to study the effect of the grinding time on the fineness and water-demand ratio of CFBA and the compressive and flexural strengths of the mortar. The results showed that the grinding process could ensure that CFBA meets the standards for second-class pulverized coal ash as a cement admixture. The main conclusions are as follows.

Grinding can effectively reduce the particle size of CFBA and ensure it meets the standard requirements. However, with the increase in grinding time, the particles are attracted by electrostatic force and van der Waals force, and the particle size of CFBA caused by agglomeration increases.Grinding can reduce the water-demand ratio of CFBA. Because the fineness of CFBFA is still relatively coarse and the carbon content of CFBFA is higher, grinding only CFBBA cannot ensure that CFBA meets the water-demand ratio requirement. When the grinding time of CFBBA is 10 min and the grinding time of CFBFA is 4 min, the water-demand ratio of CFB meets the requirements.Grinding can improve the compressive and flexural strengths of the mortar when CFBA is used as a cement admixture. Under the condition that the grinding time of CFBFA is 4 min, when the grinding time of CFBBA is 10 min rather than 2 min, the compressive strength of the 28-day-old mortar test block of CFBA increases by 7.05% and the flexural strength changes slightly. The strength activity coefficient of CFBA at 28 d meets the requirements of cement admixtures.CFBA can be used as a cement admixture to meet the requirements of the GB/T 1596-2017 standard by grinding. The optimal grinding time of CFBBA is 10 min and the optimal grinding time of CFBFA is 4 min.

## Figures and Tables

**Figure 1 materials-15-05610-f001:**
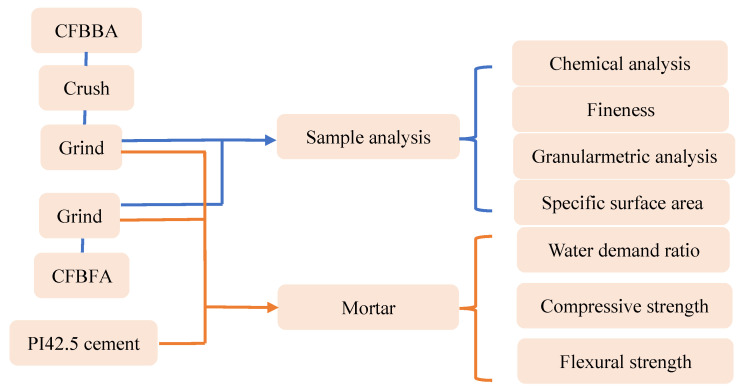
Experimental process.

**Figure 2 materials-15-05610-f002:**
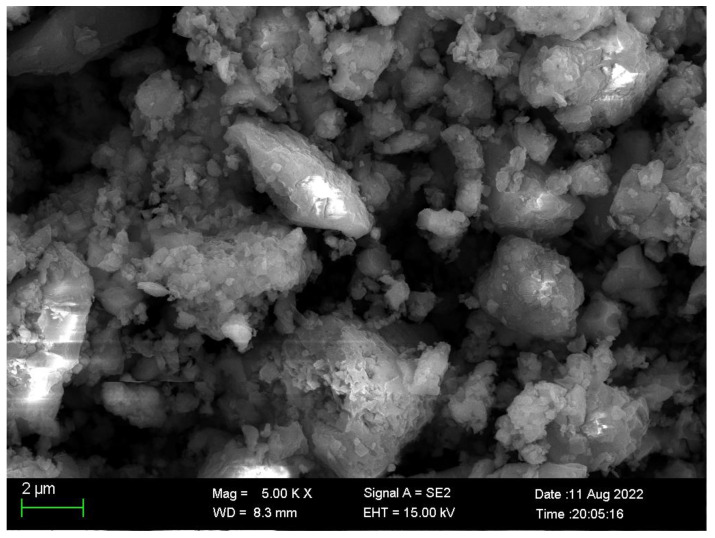
SEM images of CFBA.

**Figure 3 materials-15-05610-f003:**
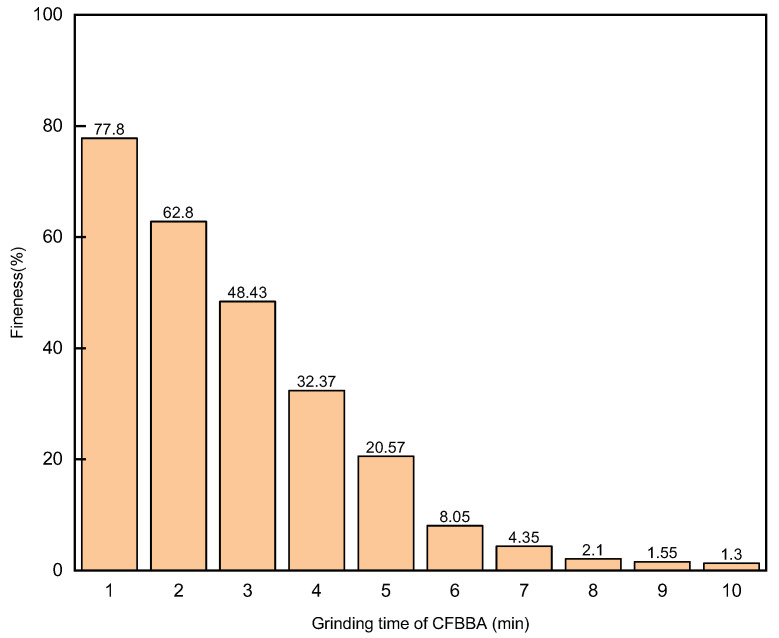
Effect of grinding time on the fineness of CFBBA.

**Figure 4 materials-15-05610-f004:**
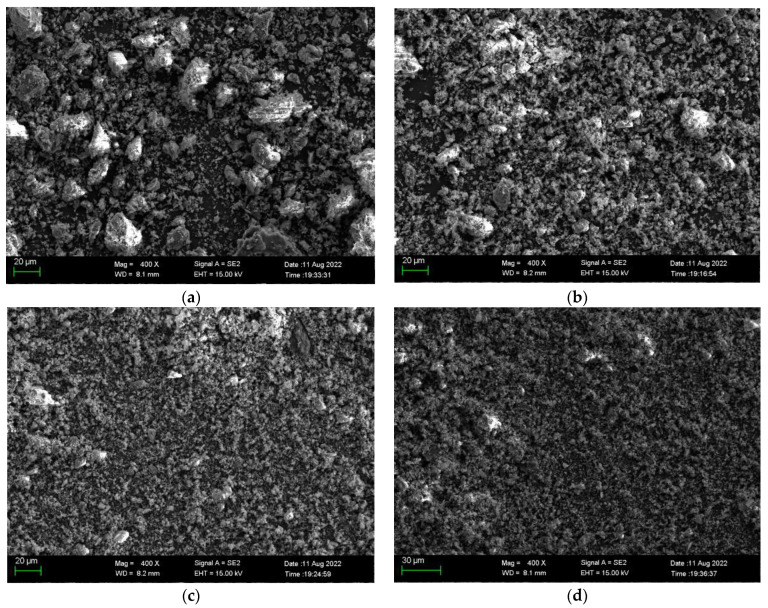
SEM images of CFBA with different grinding times: (**a**) 4 min, (**b**) 8 min, (**c**) 16 min, and (**d**) 20 min.

**Figure 5 materials-15-05610-f005:**
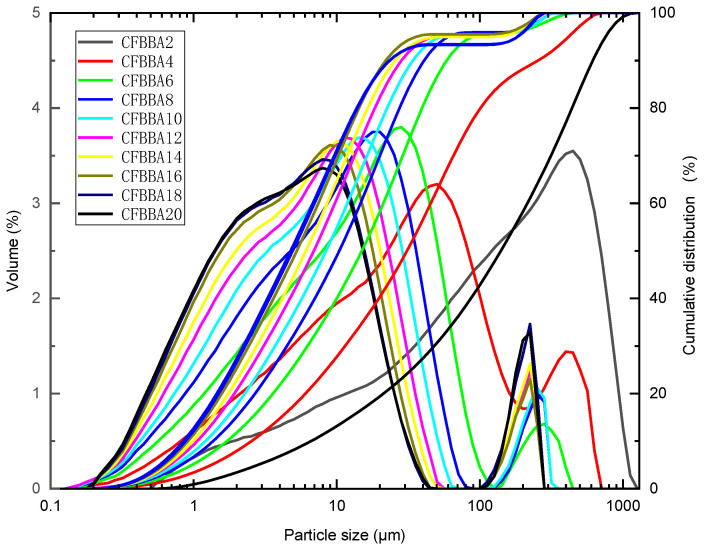
Effect of grinding time on the particle size distribution of CFBBA.

**Figure 6 materials-15-05610-f006:**
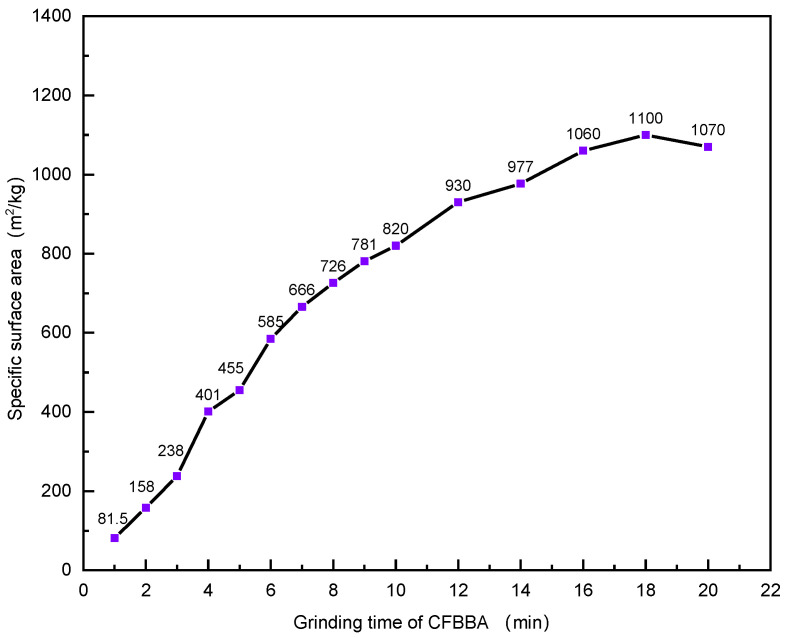
Effect of grinding time on the specific surface area of CFBBA.

**Figure 7 materials-15-05610-f007:**
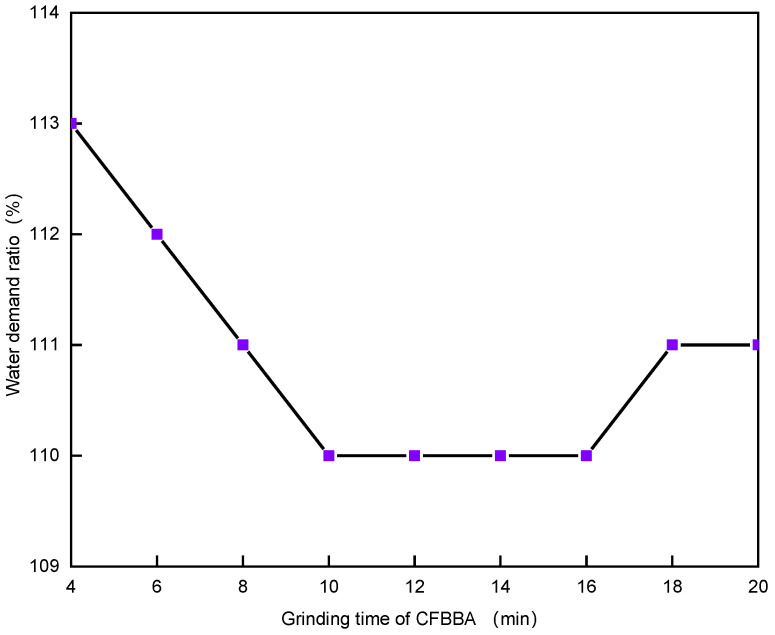
Effect of CFBBA grinding time on the water-demand ratio of CFBA.

**Figure 8 materials-15-05610-f008:**
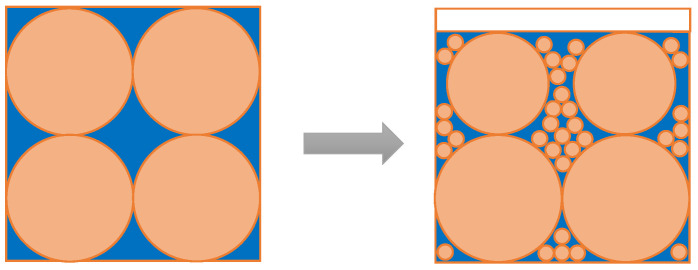
Effect of particle porosity of CFBA on water-demand ratio.

**Figure 9 materials-15-05610-f009:**
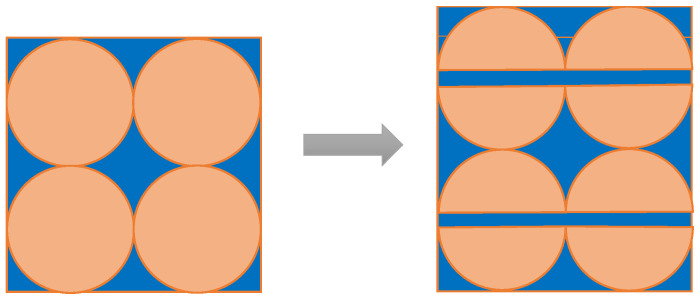
Effect of the specific surface area of CFBA on water-demand ratio.

**Figure 10 materials-15-05610-f010:**
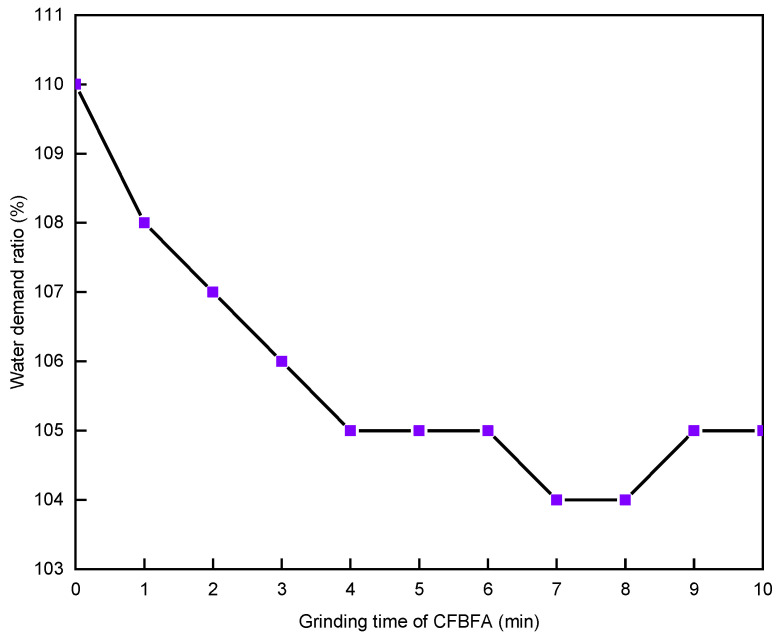
Effect of CFBFA grinding time on the water-demand ratio of CFBA with a ratio of CFBFA to CFBBA of 7:3.

**Figure 11 materials-15-05610-f011:**
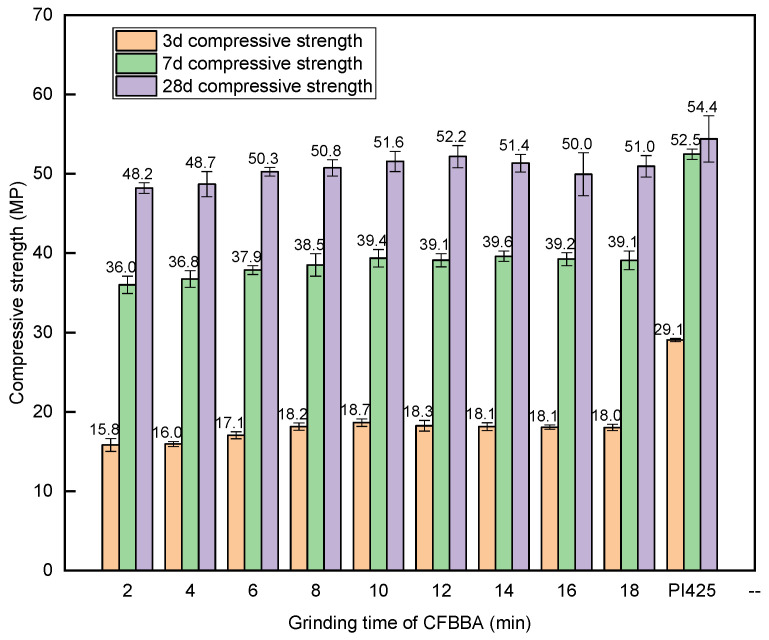
Compressive strength of mortar with different grinding times of CFBBA.

**Figure 12 materials-15-05610-f012:**
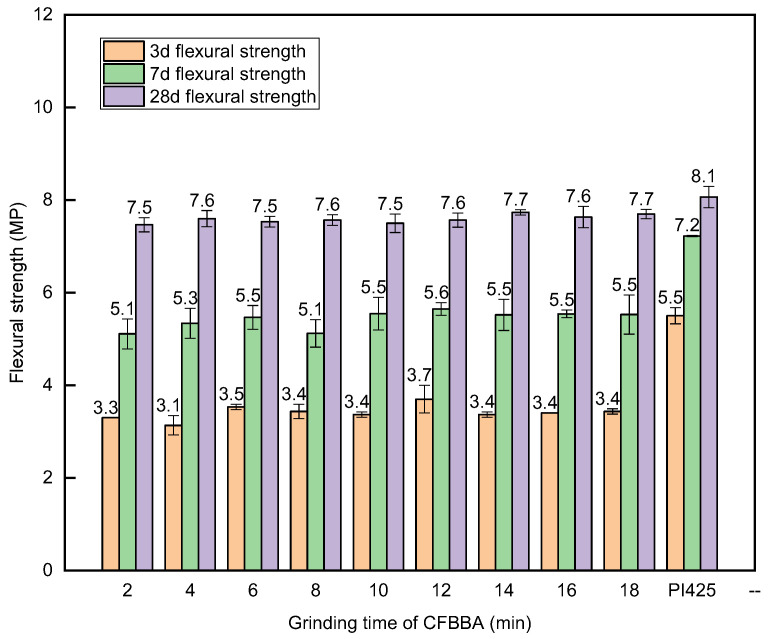
Flexural strength of mortar with different grinding times of CFBBA.

**Figure 13 materials-15-05610-f013:**
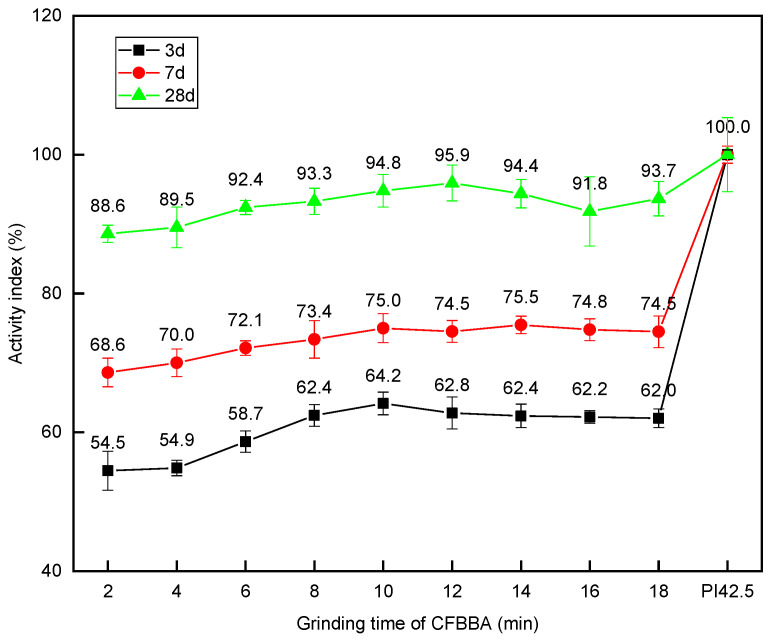
Activity index of CFBBA mortar with different grinding times.

**Table 1 materials-15-05610-t001:** The composition of mortar used for testing the water-demand ratio of CFBA.

Type of Mortar	PI42.5 Cement/g	CFBBA/g	CFBFA/g	Sand/g	Deionized Water/g
Contrast mortar	250	-	-	750	W1
Test mortar	175	52.5	22.5	750	W2

**Table 2 materials-15-05610-t002:** The composition of mortar used for testing strength of CFBA.

Type of Mortar	PI42.5 Cement/g	CFBBA/g	CFBFA/g	Sand/g	Deionized Water/g
Contrast mortar	450	-	-	1350	225
Test mortar	315	94.5	40.5	1350	225

**Table 3 materials-15-05610-t003:** Chemical composition of CFB ash.

Compound	SiO_2_	Al_2_O_3_	Fe_2_O_3_	CaO	TiO_2_	MgO	K_2_O	Na_2_O	SO_3_	Other
CFBFA	45.42	24.07	12.17	6.14	5.63	2.37	1.69	0.542	0.505	1.463
CFBBA	41.84	24.29	17.17	4.71	6.32	1.51	1.65	0.569	0.450	1.651

**Table 4 materials-15-05610-t004:** Comparison of physical and chemical characteristics of CFBA with Chinese standards.

Project	Second-Class Pulverized Coal Ash Standard	CFBFA	CFBBA
Fineness	≤30.0	15.8	100
Water-demand ratio/%	≤105	110	-
LOI/%	≤8.0	2.56	1.17
Moisture content/%	≤1.0	0.30	0.31
SO_3_ mass fractions/%	≤3.0	0.505	0.450
free CaO mass fractions/%	≤1.0	0.22	0.41
SiO_2_, Al_2_O_3_, and Fe_2_O_3_ total mass Fractions/%	≥70.0	81.66	83.30
Density/(g/cm^3^)	≤2.6	2.44	2.59
Activity index/%	≥70.0	-	-

**Table 5 materials-15-05610-t005:** Main data of particle size distribution of CFBBA with different grinding times.

Sample	d(0.1)/μm	d(0.5)/μm	d(0.9)/μm	Sample	d(0.1)/μm	d(0.5)/μm	d(0.9)/μm
CFBBA1	14.992	226.655	615.847	CFBBA9	1.268	9.194	35.649
CFBBA2	6.612	140.303	615.358	CFBBA10	1.207	8.42	33.960
CFBBA3	3.865	62.734	513.686	CFBBA12	1.058	6.914	28.272
CFBBA4	2.534	30.974	254.954	CFBBA14	0.986	6.084	25.524
CFBBA5	2.205	24.383	240.411	CFBBA16	0.913	5.26	21.982
CFBBA6	1.710	14.972	58.471	CFBBA18	0.878	4.92	23.416
CFBBA7	1.498	11.967	45.674	CFBBA20	0.878	5.687	28.919
CFBBA8	1.368	10.413	40.727				

## Data Availability

Not applicable.

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
