# Peer review of "Feasibility Study of Grinding Circulating Fluidized Bed Ash as Cement Admixture"

_materials, 2022, doi:10.3390/ma15165610_

Round 1
Reviewer 1 Report
Reviews
In the review of research manuscript titled: “Feasibility Study of Grinding Circulating Fluidized Bed Ash as Cement Admixture”, the article is written very well and explanation of the results is also good. I would like to see this article publish but some minor modifications are required.
1- Introduction part is very short. Can you please cite some more articles to relate it with your work highlighting the importance of the limestone being fed to the furnace during the CFB boiler operation in order to control the SO2 emission, which results in relatively high content of sulfur and free-CaO in the ash up to one more paragraph?
2- As claimed the particle size is reduced to fix the water usage issue, please provide some SEM images and compare them before and after grinding.
3- Figure 3. Effect of grinding time on the particle size distribution of CFBBA, please provide some additional results to support this claim.
4- Figure 5. Effect of CFBBA grinding time on the water demand ratio of CFBA. Please compare three to four samples in one graph with different grinding time, different particle size for the water demand ratio to analyze it deeply.
5-Please add some more references and cite good articles to emphasis on the size of different particles effecting the amount of water? Is that possible to gain a level of the size of particle so small that no amount of water is needed at all.

Reviewer 2 Report
The results are presented clearly. However, the novelty is lacking. Some comments from the reader are as follows:
1. Introduction should be significantly improved. The research problem is a technical problem. What is the novelty of this study?
2. Please add more pictures to describe the materials used. Significantly, the lack of SEM results proves the size particle size after grinding.
3. How can the author determine the fineness of CFBBA and the particle size distribution of CFBBA.
4. BA1, BA2, BA3,..., BA20 should be noted.
5. All testing procedures should be described in detailed steps.
6. How to determine the water demand ratio? Please clarify
7. Line 217-220: "It can be seen that the water demand ratio of 217 CFBA first decreases and then increases with the increasing of grinding time of CFBBA, 218 but it still does not reach the 105% requirement in GB/T 1596-2017, so another experiment 219 was carried out." Please explain the reason of these results
8. Obtained results must be explained in more detail.
8. If possible, the authors do some more tests related to microstructure and durability.
Reviewer 3 Report
General comments:
Abstract:
1. In the Abstract for a better understanding of the main material of the study (CFBA), it is suggested that the authors provide a mention of its origin to clarify this from the beginning and not during the reading of the manuscript. “CFB ash is the combustion product of coal at 850℃-950 ℃, and the characteristics of CFBA are usually loose and porous in structure with many amorphous substances.” this is mentioned only in the middle of the introduction.
2. In the Abstract it is mentioned how grinding affects the water requirement of 2 materials (CFBBA and CFBFA) but there is no comparison with the main material (CFBA) nor are the results mentioned in it. In addition, it is mentioned how grinding contributes to the compressive strength on day 28 of hardening of CFBA, and the other two materials are not compared or mentioned, despite having been talking about the 3 in that paragraph.
Introduction:
3. The introduction is adequate, it talks about the national context and about the environmental context of the country and planet, also about the lack of research on this material, that is, the present opportunity and problem are understood.
4. Check this sentence: “Grinding can reduce the problem of particle size and water demand ratio of CFBA, and improve the strength of the mortar to meet Chinese standards as a cement admixture. However, the key influencing factors on the quality of ground ash as cement admixture are still needed to investigate.” It is understood that the work to be carried out in this investigation is to study what factors influence the quality of CFBA and the hypothesis is that the material is suitable as a replacement, however, this is not completely clear.
Methodology:
5. The explanation of what was wanted to be measured needs to be improved. As it is, some tests are not repeatable. Provide the number of samples tested on each test.
6. The authors must explain clearly how the effects of CBBA and CFBFA combined, using different percentages of replacements for each one, can be identified in the results. Results from mixtures using only CBBA and CFBFA separately will provide robustness to the presented results.
7. In the conclusions he mentions the CFBA set and not the materials by themselves (CFBFA and CFBBA), and in the results, he was changing materials and tests without mentioning much, and they conclude by mentioning the unspecified set.
Results:
8. Figure 5. Please explain why water demand has a minimum within 4 and 18 minutes of griding.
9. Figures 5 and 6 present results of water demand using two different methods. Both methods are not clearly identified and described in the methodology section.
10. Figures 9, 10, and 11. Provide statistical deviation of the results.
Conclusions:
Do not use bullets to address conclusions.
Round 2
Reviewer 2 Report
Great! Accepted as its present form
Author Response
The authors gratefully acknowledge the reviewer's suggestion of the manuscript and have chosen to accept the revision.